# Clinical and Genetic Studies of the First Monozygotic Twins with Pfeiffer Syndrome

**DOI:** 10.3390/genes13101850

**Published:** 2022-10-13

**Authors:** Piranit N. Kantaputra, Salita Angkurawaranon, Krit Khwanngern, Chumpol Ngamphiw, Worrachet Intachai, Ploy Adisornkanj, Sissades Tongsima, Bjorn Olsen, Nuntigar Sonsuwan, Kamornwan Katanyuwong

**Affiliations:** 1Division of Pediatric Dentistry, Department of Orthodontics and Pediatric Dentistry, Faculty of Dentistry, Chiang Mai University, Chiang Mai 50200, Thailand; 2Center of Excellence in Medical Genetics Research, Faculty of Dentistry, Chiang Mai University, Chiang Mai 50200, Thailand; 3Division of Diagnostic Radiology, Department of Radiology, Faculty of Medicine, Chiang Mai University, Chiang Mai 50200, Thailand; 4Division of Plastic Surgery, Department of Surgery, Chiang Mai University, Chiang Mai 50200, Thailand; 5National Biobank of Thailand, National Science and Technology Development Agency, Khlong Luang, Pathum Thani 12120, Thailand; 6Dental Department, Sawang Daen Din Crown Prince Hospital, Sakon Nakhon 47110, Thailand; 7Department of Developmental Biology, Harvard School of Dental Medicine, Harvard University, Boston, MA 02115, USA; 8Department of Otolaryngology, Faculty of Medicine, Chiang Mai University, Chiang Mai 50200, Thailand; 9Division of Neurology, Department of Pediatrics, Faculty of Medicine, Chiang Mai University, Chiang Mai 50200, Thailand

**Keywords:** Pfeiffer syndrome, twins, lambdoid synostosis, discordance, hypertrophy of turbinates

## Abstract

*Objective:* To report the clinical and radiographic findings and molecular etiology of the first monozygotic twins affected with Pfeiffer syndrome. *Methods:* Clinical and radiographic examination and whole exome sequencing were performed on two monozygotic twins with Pfeiffer syndrome. *Results:* An acceptor splice site mutation in *FGFR2* (c.940-2A>G) was detected in both twins. The father and both twins shared the same haplotype, indicating that the mutant allele was from their father’s chromosome who suffered severe upper airway obstruction and subsequent obstructive sleep apnea. Hypertrophy of nasal turbinates appears to be a newly recognized finding of Pfeiffer syndrome. Increased intracranial pressure in both twins were corrected early by fronto-orbital advancement with skull expansion and open osteotomy, in order to prevent the more severe consequences of increased intracranial pressure, including hydrocephalus, the bulging of the anterior fontanelle, and the diastasis of suture. *Conclusions:* Both twins carried a *FGFR2* mutation and were discordant for lambdoid synostosis. Midface hypoplasia, narrow nasal cavities, and hypertrophic nasal turbinates resulted in severe upper airway obstruction and subsequent obstructive sleep apnea in both twins. Hypertrophy of the nasal turbinates appears to be a newly recognized finding of Pfeiffer syndrome. Fronto-orbital advancement with skull expansion and open osteotomy was performed to treat increased intracranial pressure in both twins. This is the first report of monozygotic twins with Pfeiffer syndrome.

## 1. Introduction

Pfeiffer syndrome (MIM 101600) is characterized by craniosynostosis, midface hypoplasia, broad thumbs and great toes, brachydactyly, and syndactyly. Synostosis of the coronal suture is the most common craniosynostosis in patients with Pfeiffer syndrome. Ocular hypertelorism, down-slanting palpebral fissures, strabismus, and ocular proptosis are common. Mode of inheritance is autosomal dominant with complete penetrance and phenotypic variability [1]. Mutations in *FGFR2* have been reported to be associated with most cases of Pfeiffer syndrome (95%). However, *FGFR1* mutations (5%) have also been reported in patients with a milder Pfeiffer syndrome phenotype [2,3,4]. Here, we report on the first monozygotic twins with Pfeiffer syndrome who carried an acceptor splice site mutation in *FGFR2* and were discordant for lambdoid synostosis. Midface hypoplasia, narrow nasal cavities, and hypertrophic nasal turbinates were the causes of severe upper airway obstruction and subsequent obstructive sleep apnea. Increased intracranial pressure in both twins were corrected by fronto-orbital advancement with skull expansion and open osteotomy. Hypertrophy of turbinates found in these twins appears to be the newly recognized finding of Pfeiffer syndrome.

## 2. Patient Report

Twins A (Figure 1) and B (Figure 2) were products of healthy non-consanguineous Thai parents. Their father and mother were 26 and 25 years old at the time of conception, respectively. Their mother took oral contraceptive medication until four weeks after absence of menstruation. No family history of craniosynostosis or congenital birth defects was evident. The twins were born at term by normal delivery at 39 weeks and 3 days of gestation. Both of them had good Apgar scores of 9 and 10 at 1 and 5 min, respectively. Birthweight, birth length, and OFC of twin A were 2,950 g (10 centile), 49 cm (25 centile), and 34 cm (50 centile), respectively. Birthweight, birth length, and OFC of twin B were 2,450 (<3 centile), 45 cm (<3 centile), and 35 cm (50 centile), respectively. Clinically, both twins had brachycephaly, ocular proptosis, broad thumbs and great toes, and double hair whorls on the top of their heads (Figure 1A–D and Figure 2A–D). Both of them received phototherapy for the treatment of neonatal jaundice. They started having respiratory tract infections followed by respiratory failure at age around one month old. Neurological examinations revealed good eye contact with social smiles. At age six months, gross and fine motor developments were delayed. Both twins could sit without support but were unable to crawl. Bulging of anterior fontanelle, a clinical sign of increased intracranial pressure, was not observed. Ophthalmologic examination showed no signs of papilledema, an optic disc swelling with blurred margin, which is a sign of chronic increased intracranial pressure [5,6].

Both twins suffered severe upper airway obstruction. They frequently had snoring and obstructive sleep apnea. The lowest oxygen saturation of nocturnal pulse oximetry tests of twins A and B were 66% and 80%, respectively. Sleep apnea tests concluded that twins A and B had severe and moderate sleep apnea, respectively. Tracheostomy was performed in both twins at the age of nine months.

Radiographic and computed tomography of both twins at age 6 months showed bicoronal synostosis (Figure 3A,B), bilateral dehiscence of posterior semicircular canals (Figure 4C,D), shallow orbit leading to exorbitism (Figure 4E,F), midface hypoplasia (Figure 4G,H), and short anterior cranial base (Figure 5C,D). Twin A had left lambdoid synostosis (Figure 3C) and effusion of the right middle ear cavity and mastoid air cells (Figure 4A). Twin B had bilateral effusion of middle ear cavities and mastoid air cells (Figure 4B). The lambdoid sutures of twin B appeared patent (Figure 3D).

Both twins had unremarkable naso- and oropharyngeal spaces (Figure 5A,B). Narrowness of nasal cavities and hypertrophy of superior, middle, and inferior turbinates were observed in both twins (Figure 6A–D). Craniolacunia was observed in the parietal and occipital bones of both twins (Figure 3C–D). Diffuse thumbprinting phenomenon or beaten copper appearance on skull radiographs and widening of sagittal sutures were observed in both twins, indicating increased intracranial pressure. The other radiographic signs of increased intracranial pressures such as erosion of dorsum sellae and suture diastasis were not evident [7]. As a result of increased intracranial pressure, fronto-orbital advancement with skull expansion was performed by a plastic surgeon (K.K.) in both twins at the ages of 6 and 8 months, respectively. Open osteotomy was performed for coronal synostosis correction and fixation with resorbable plate osteosynthesis. Subsequently fronto-orbital bars were anteriorly repositioned for 15 mm. Both twins had periodic follow-ups to be evaluated for increased intracranial pressure and secondary craniosynostosis.

## 3. Results

### 3.1. Whole Exome and Sanger Direct Sequencing and Detection of the Parental Origin of the Mutant Allele

Whole exome sequencing showed a *de novo* heterozygous base substitution NM_000141.4: c.940-2A> G (rs1057519041) in intron 8 at the 3′ splice acceptor site of the *FGFR2* gene, leading to abnormal IIIc acceptor splicing in both twins. Sanger sequencing confirmed the mutation (Figure 7). This mutation is known to be pathogenic for Pfeiffer syndrome. In order to search for the parental origin of the mutant allele, we observed neighboring *FGFR2* variants around NM_000141.4; (*FGFR2*): c.940-2A>G and found two variants located downstream of this mutation, namely, rs11199991 (NM_000141.4: c.1287 + 4293G>C, 6642 bp from the mutation) and a novel intron variant (NM_000141.4: c.1287 + 4329_1287 + 4338dup, 6688 bp from the mutation), with different genotypes in both parents. For NM_000141.4: c.1287 + 4329_1287 + 4338dup, the father and the twins shared the ATTTAT/ATTTATTTTAT genotype, while the mother had the ATTTATTTTATTTTAT/ATTTATTTTAT genotype. For rs11199991, the father and the twins also shared CC genotypes (homozygous wild-type), while the mother’s genotype was CG (heterozygous). The father and both twins shared the same genotypes in both NM_000141.4:c.1287 + 4293G>C and NM_000141.4:c.1287 + 4329_1287+4338dup, suggesting that the shared alleles are in the same haplotype. This shows that the mutant allele was from their father’s chromosome.

### 3.2. Hearing Assessment

Auditory brainstem response (ABR) test of the right ear of twin A demonstrated an air conduction threshold at 45 dB (normal = 20 dB) and bone conduction threshold at 20dB, with an air–bone gap of 25 dB, indicative of conductive hearing loss in the right ear. The ABR test of the left ear was unremarkable. Auditory brainstem test of the right ear of twin B demonstrated threshold of air conduction at 65 dB and of bone conduction at 25 dB with an air–bone gap of 40 dB (normal = 20 dB), indicative of conductive hearing loss in the right ear. The hearing on the left was normal with ABR threshold 25 dB. Tympanograms of both patients showed negative results in both ears at both 226 and 1000 Hz, indicative of middle ear effusion.

## 4. Discussion

We report the first monozygotic twins affected with Pfeiffer syndrome with a heterozygous *FGFR2* mutation (NM_000141.4: c.940-2A>G). Both twins had diffuse beaten copper appearance on the skull radiograph, a sign of increased intracranial pressure. Surgical intervention was performed quite early in both twins to prevent them from having papilledema, which may lead to irreversible vision loss, bulging of anterior fontanelle, and hydrocephalus [5], as well as to maximize therapeutic outcomes because infants are able to ossify small cranial defects, thus minimizing the need for bone grafting [8]. Periodic follow-ups after surgery for clinical manifestations of increased intracranial pressure were performed because of the high rate of relapse and secondary synostosis in patients with syndromic craniosynostosis [9], which is most likely the result of phenotypic effects of *FGFR* mutations, tissue-specific expression of different FGFR isoforms, other modifying factors, and possibly environmental factors [10].

### 4.1. FGFR2 Mutations and Pfeiffer Syndrome

We report the first monozygotic twins affected with Pfeiffer syndrome who carried a *FGFR2* mutation and were discordant for lambdoid synostosis. The 3′ acceptor splice site mutation c.940-2A>G in *FGFR2* found in our patients has been reported in 16 patients. They all had type 1 Pfeiffer syndrome, but none of them had lambdoid synostosis [2,3,11,12,13,14,15]. Twins with craniosynostosis, midface hypoplasia, and exorbitism were previously reported to have Pfeiffer syndrome [16]. Unfortunately, the pictures of the broad thumbs and great toes, which are the characteristic features of Pfeiffer syndrome, were not illustrated. The authors did not report on the causative mutation and the facial features of patients resembling those of Crouzon syndrome [16]. Both of our patients had craniolacunia in the parietal and occipital bones. Craniolacunia is characterized by round or irregular gaps in the skull, covered by a diaphragm of periosteum and dura mater, bound by bony ridges. It is caused by periosteal dysplasia and is considered the effect of *FGFR2* mutations [17].

Fibroblast growth factors, a family of intercellular-signaling molecules, transduce their signals by activating specific cell surface tyrosine kinase receptors, FGFRs. FGF/FGFR signaling depends on dimerization of receptor molecules brought together by ligand binding in cooperation with heparan sulfate proteoglycans. FGFRs differ from each other in their ligand affinities and tissue distribution [18]. In humans, *FGFR* genes encode an extracellular domain composed of two or three immunoglobulin-like (Ig) domains, a transmembrane segment, and a cytoplasmic tyrosine kinase domain [18]. Ligand binding involves IgII and IgIII domains, but binding specificity is determined only by the IgIII domain [19]. In mice, heterozygous abrogation of Fgfr2 exon 9 (IIIc) causes splicing switch, resulting in a gain-of-function mutation, upregulated *FGFR2* signaling, precocious ossification of cranial sutures, and subsequent craniosynostosis [18].

As was found in our patients, *FGFR2* mutations are most commonly located in the IgIII domain and encoded by exons 8 (IIIa) or 10 (IIIc) [20,21,22]. Mutations in *FGFR2* have been shown to cause a wide range of genetic disorders, including Pfeiffer, Crouzon, Apert, Antley–Bixler, Beare–Stevenson cutis gyrata, Jackson–Weiss, bent bone dysplasia, and Seathre–Chotzen-like syndromes. The same mutations in *FGFR2* can give rise to Crouzon, Jackson–Wei, and Pfeiffer syndromes. It is likely that other modifying genetic factors or epigenetic factors have an influence on abnormal gain of function, upregulated FGF signaling, and subsequent phenotypic variability [2,20,23,24,25]. It is noteworthy that the c.940-2A>G mutation found in our patients has been reported 16 times and is always associated with Pfeiffer syndrome [2,3,11,12,13,14,15,26,27].

### 4.2. FGFR2 Mutation and Its Parental Origin

Spontaneous mutations in *FGFR2* were usually of paternal origin and associated with advanced paternal age. This is because FGF/FGFR signaling has a crucial role in initiation, maintenance of spermatogenesis, and clonal expansion of spermatogonial cells expressing proteins with gain-of-function or oncogenic properties [21,28,29]. The father and mother of our patients were 26 and 25 years old, respectively, at the time of conception. The mutations found in our patients were of paternal origin, suggesting that the mutational event could have taken place during spermatogenesis [28], even though the father was quite young.

### 4.3. Malformation of Middle Ear and Hearing Loss

Otological malformations and hearing loss are common features of Pfeiffer syndrome. Generally, patients with a mild form of Pfeiffer syndrome have normal ear anatomy and hearing [30]. On the right ears of both twins, effusion of the middle ears and mastoid air cells was observed. On the left side, it was observed only in twin B. Effusion in the mastoid air cells is secondary to effusion of the middle ear. The middle ear effusion in our patients, which is common in patients with Pfeiffer syndrome, is caused by the narrowness of the eustachian tube secondary to the malformations of cranial base and nasopharynx [30,31,32]. Long-term middle ear effusion can lead to conductive hearing loss [33]. It also may impair postural stability, vestibulo-spinal reflex, and motor development of the patients [34].

### 4.4. Dehiscence of Posterior Semicircular Canals

Dehiscence of posterior semicircular canals of our patients has also been reported in patients with Pfeiffer syndrome [35,36]. The posterior semicircular canal is located caudally in the body of the petrous part of the temporal bone. The posterior branch is likely to have dehiscence of the thin overlying temporal bone facing the posterior fossa [37]. This dehiscence or opening in the bone, overlying the posterior semicircular canal, creates a window into the inner ear that allows the canal to be responsive to sound and to changes in pressure in the membranous labyrinth, causing vertigo [38]. However, it can also be asymptomatic, as occurred in our patients. FGF signaling through Fgfr2 (IIIb) has important roles in the development of the outer sulcus, stria vascularis, and spiral prominence of cochlea and otic induction. Its expression is robust and consistent in presumptive stria vascularis and spiral prominence [19]. *FGFR2 (IIIb)* mRNA is expressed in the non-sensory epithelium of the otocyst that develops to structures such as endolymphatic and semicircular ducts. *FGFR2* appears to be crucial for the development of the inner ear.

### 4.5. Increased Intracranial Pressure, Upper Airway Obstruction, and Sleep Apnea

Both twins suffered moderate and severe sleep apnea as a result of narrow nasal cavities and hypertrophic turbinates, leading to upper airway obstruction. They had experienced a few episodes of sleep apnea. Tracheostomy was performed in both patients at the age of nine months. The prevalence of obstructive sleep apnea among patients with craniosynostosis syndromes is highest in patients with Apert syndrome (80.6%), followed by Pfeiffer (72.7%), Crouzon with acanthosis nigricans (66.7%), and Crouzon (64.7%) syndromes. This is the result of a markedly short anterior cranial base and a severely hypoplastic and retrognathic midface, leading to narrowness of the nasal cavity and subsequent upper airway obstruction [31,39]. Hypertrophy of turbinates, which reduces the size of the nasal air passage, has never been reported in patients with Pfeiffer syndrome. Surgical intervention was performed early in both twins because obstructive sleep apnea and increased intracranial pressure have been reported to cause optic neuropathy in patients with craniosynostosis. Therefore, early detection and treatment of increased intracranial pressure are crucial in order to prevent irreversible loss of vision [40].

### 4.6. Discordance in Monozygotic Twins

Monozygotic twins result from the division of the single embryo at the two-cell stage into two independent daughter cells, and are expected to have identical genomes, with few exceptions. Therefore, the phenotypic difference between monozygotic twins, such as the lambdoid synostosis in twin A, is considered to be the combined effect of environmental and nongenetic factors [41]. Each monozygotic twin is not exposed to identical intrauterine factors and maternal variables. Epigenetic factors such as DNA methylation and histone modification regulate heritable states of gene expression. This epigenome varies from tissue to tissue, controlling differential gene expression and providing specific identity to each cell type [41]. Regarding discordance for lambdoid synostosis, *FGFR2* mutations predispose the patients to craniosynostoses. Somatic mutations and intrauterine elements may have caused genetic discordance in our patients. In addition, deformation as a result of intrauterine constraint of twin pregnancy may have influenced the discordance in twins as well [41,42].

## 5. Materials and Methods

### 5.1. Whole Exome Sequencing, Sanger Sequencing, and Bioinformatic Analysis

Whole exome sequencing with the Agilent SureSelect V6 exome capture kit and Sanger direct sequencing were performed on the patients and their unaffected parents.

### 5.2. Hearing Assessment

An auditory brainstem response (ABR) test was performed on both patients.

## 6. Conclusions

We report on monozygotic twins affected with Pfeiffer syndrome and carrying a *FGFR2* mutation. Both twins were discordant for lambdoid synostosis. Midface hypoplasia, narrow nasal cavities, and hypertrophic nasal turbinates resulted in severe upper airway obstruction and subsequent obstructive sleep apnea in both twins. Hypertrophy of the nasal turbinates appears to be a newly recognized finding of Pfeiffer syndrome. Increased intracranial pressure in both twins were corrected early by fronto-orbital advancement with skull expansion and open osteotomy, in order to prevent the more severe consequences of increased intracranial pressure, including hydrocephalus, bulging of anterior fontanelle, papilledema, erosion of dorsum sellae, and diastasis of suture.

## Figures and Tables

**Figure 1 genes-13-01850-f001:**
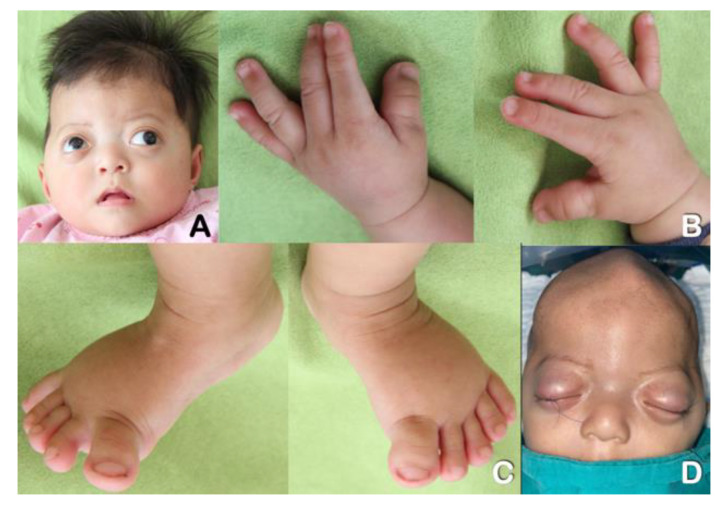
**Twin A at age 6 months.** (**A**) Brachydactyly, ocular proptosis, and depressed nasal bridge. (**B**) Brachydactyly and broad thumbs. (**C**) Broad great toes. (**D**) Preoperative view of the cranium. Note ridges at metopic and sagittal sutures as a result of synostosis.

**Figure 2 genes-13-01850-f002:**
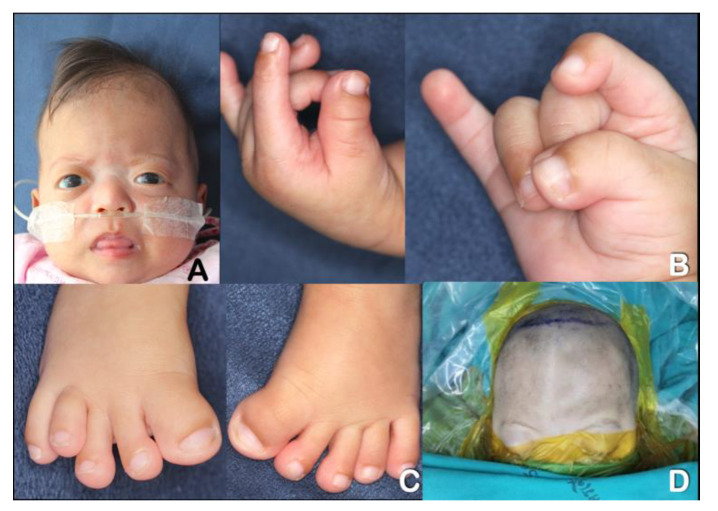
**Twin B at age 3.5 months.** (**A**) Brachydactyly, ocular proptosis, and depressed nasal bridge. (**B**) Brachydactyly and broad thumbs. (**C**) Broad great toes. (**D**) Preoperative view of the cranium. Note ridge at metopic suture as a result of synostosis.

**Figure 3 genes-13-01850-f003:**
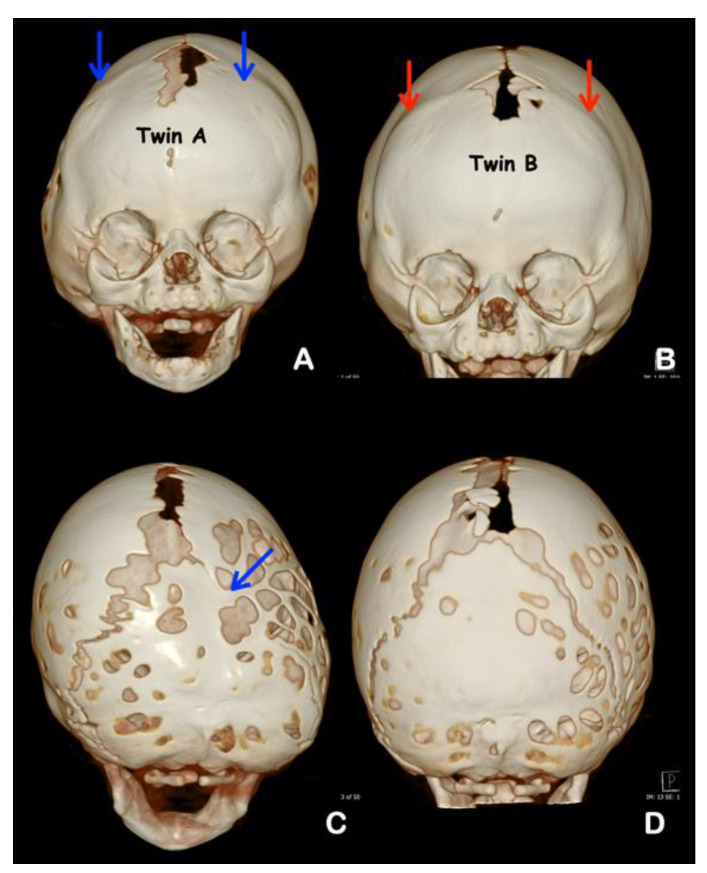
**Radiographs and 3D CT reconstruction of twins A (A,C) and B (B,D).** (**A**) Coronal synostosis (blue arrows) of twin A. (**B**) Coronal synostosis (red arrows) of twin B. (**C**) Note the craniolacunia of twin A. Lambdoid synostosis (blue arrow) and ipsilateral mastoid and contralateral parietal bossing with associated downward tilt of the cranial base toward the affected side. (**D**) Craniolacunia and patent lambdoid suture of twin B.

**Figure 4 genes-13-01850-f004:**
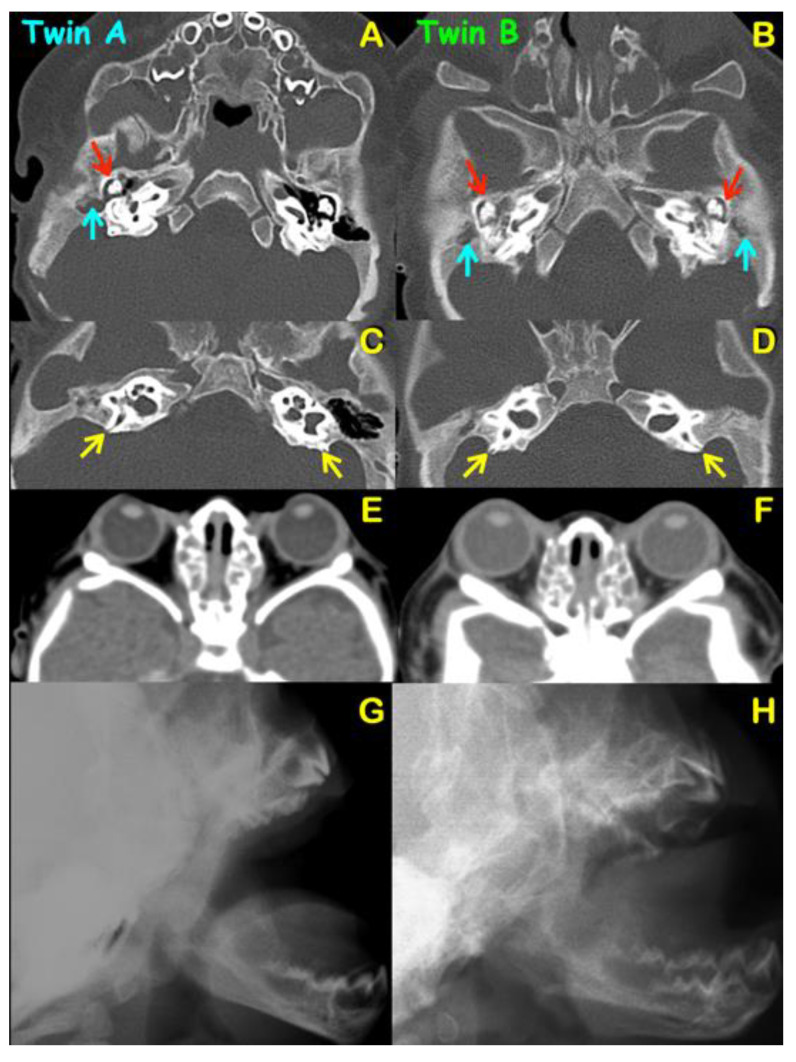
**Axial CT scan and lateral skull radiographs of twins A (A,C,E,G) and B (B,D,F,H).** (**A**) Twin A. Effusion of right middle ear cavity (red arrow) and mastoid air cells (blue arrow). (**B**) Twin B. Bilateral effusion of middle ear cavities (red arrows) and mastoid air cells (blue arrows). (**E**,**F**) CT scan of brain. Shallow orbits leading to exorbitism of (**E**) twin A and (**F**) twin B. (**G**,**H**) Lateral skull radiographs demonstrating maxillary hypoplasia of (**G**) twin A. and (**H**) twin B.

**Figure 5 genes-13-01850-f005:**
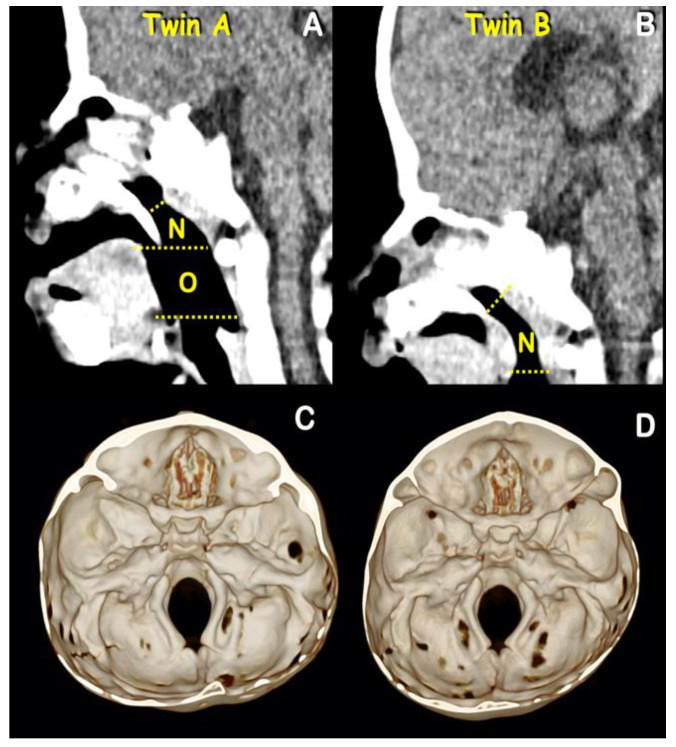
**Sagittal view CT.** (**A**) Normal naso-and oropharyngeal airways of twin A. (**B**) Normal naso-and oropharyngeal airways of twin B. **Three-dimensional CT reconstruction**. (**C**) Short anterior cranial bases of twin A and (**D**) twin B.

**Figure 6 genes-13-01850-f006:**
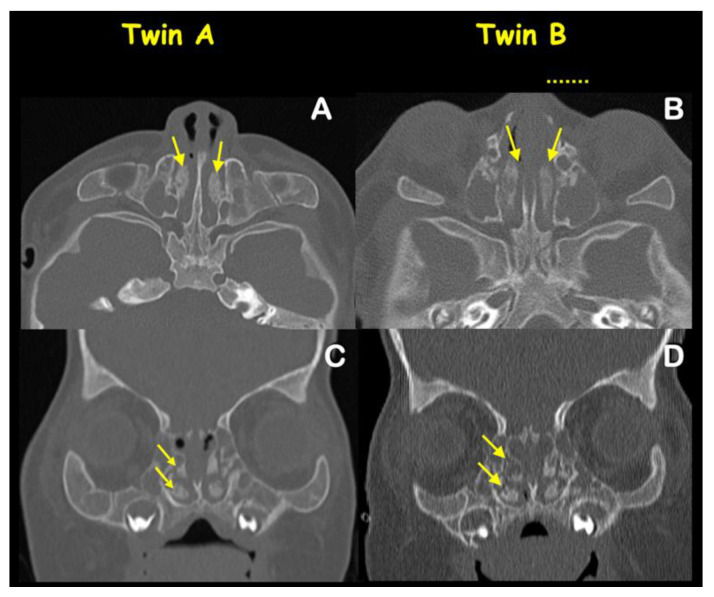
**Axial and coronal views CT (bone window) of****twins A and B.** Obstruction of bilateral nasal cavities (arrows) in (**A**) twin A and (**B**) twin B. Bilateral hypertrophy of turbinates (arrows) in (**C**) twin A (**D**) twin B.

**Figure 7 genes-13-01850-f007:**
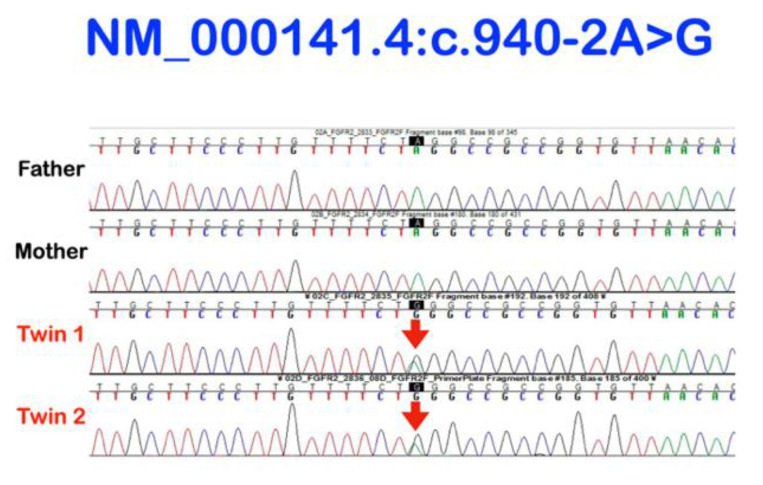
**Electropherograms of a****FGFR2****mutation.** A de novo heterozygous base substitution NM_000141.4:c.940-2A>G in intron 8 at the 3′ splice acceptor site of the FGFR2 gene, leading to abnormal IIIc acceptor splicing in both twins. The parents did not have the mutation.

## Data Availability

Not applicable.

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
