# Peer review of "Clinical and Genetic Studies of the First Monozygotic Twins with Pfeiffer Syndrome"

_genes, 2022, doi:10.3390/genes13101850_

Round 1
Reviewer 1 Report
Dr. Kantaputa and coauthors present a case of monozygotic twins with Pfeiffer syndrome caused by an FGFR2 c.940-2A>G mutation leading to altered splicing. Although this splice site variant has been described in other cases of Pfeiffer syndrome, this is the first case of monozygotic twins. The phenotypes of the twins are well described. This reviewer would suggest greater emphasis on the discordant phenotypes as this will give clues about the epigenetic influences on the phenotypes of Pfeiffer syndrome. As currently written, it adds little to our knowledge of phenotypic variability in FHFR2 medicated craniosynostosis. Focusing on the discordant phenotypes will make the manuscript much more impactful
Line 23 – Explain the purpose of the exome sequencing in the abstract (e.g. “to determine parent of origin”)
Line 34 – Wording “could correct” – “was performed to treat”
Figures 1 and 2 – What is the value of the intraoperative images? Consider deletion of 1D and 2D. The manuscript is about phenotype, not treatment.
Line 86 – Consider substituting “lowest” for “minimal”
Figure 2D – This appears to be a metopic ridge, not a sagittal ridge. If the image is kept, it would be helpful to have it rotated 180 degrees. Note, no sagittal ridge is apparent in figure 3.
Line 139 – It is not specifically stated why exome sequencing was pursued. This paragraph should be rearranged to state the purpose. It should also explicitly state why the parent of origin is important for this report.
Line 184 – The existence of another report of presumed monozygotic twins with Pfeiffer syndrome should be added to the introduction.
Line 208 – Typographical error “Weiss”
Line 260 – This sentence is unrelated to upper airway obstruction.
Author Response
Response to comments of Reviewer 1
Dr. Kantaputa and coauthors present a case of monozygotic twins with Pfeiffer syndrome caused by an FGFR2 c.940-2A>G mutation leading to altered splicing. Although this splice site variant has been described in other cases of Pfeiffer syndrome, this is the first case of monozygotic twins. The phenotypes of the twins are well described. This reviewer would suggest greater emphasis on the discordant phenotypes as this will give clues about the epigenetic influences on the phenotypes of Pfeiffer syndrome. As currently written, it adds little to our knowledge of phenotypic variability in FGFR2 medicated craniosynostosis. Focusing on the discordant phenotypes will make the manuscript much more impactful
RESPONSE
We agree with your comments that the discordant phenotype is the important finding of this paper. This has been thoroughly described in the discussion as…
Discordance in monozygotic twins
Monozygotic twins result from the division of the single embryo at the two-cell stage into two independent daughter cells, and are expected to have identical genomes, with few exceptions. Therefore, the phenotypic difference between monozygotic twins, such as the lambdoid synostosis in twin A, is considered to be the combined effect of environ-mental and nongenetic factors [41]. Each monozygotic twin is not exposed to identical intrauterine factors and maternal variables. Epigenetic factors such as DNA methylation and histone modification regulate heritable states of gene expression. This epigenome varies from tissue to tissue, controlling differential gene expression and providing specific identity to each cell type [41]. Regarding discordance for lambdoid synostosis, FGFR2 mutations predispose the patients to craniosynostoses. Somatic mutations and intrauterine elements may have caused genetic discordance in our patients. In addition, deformation as a result of intrauterine constraint of twin pregnancy may have influenced the discordance in twins as well [41,42].
Line 23 – Explain the purpose of the exome sequencing in the abstract (e.g. “to determine parent of origin”)
RESPONSE
Thank you for this comment. This has been added as …. The father and both twins shared the same haplotype, indicating that the mutant allele was from their father’s chromosome. The discussion on FGFR2mutation and its parental originis in the discussion part.
Line 34 – Wording “could correct” – “was performed to treat”
RESPONSE
Thank you for this comment. This has been corrected as …. Fronto-orbital advancement with skull expansion and open osteotomy was performed to treat increased intracranial pressure in both twins.
Figures 1 and 2 – What is the value of the intraoperative images? Consider deletion of 1D and 2D. The manuscript is about phenotype, not treatment.
RESPONSE
Thank you for this valuable comment. We have deleted the intraoperative figures 1E and 2E and 2F from the manuscript.
Line 86 – Consider substituting “lowest” for “minimal”
RESPONSE
Thank you for this comment. This has been corrected as …. Lowest oxygen saturation of nocturnal pulse oximetry tests of twins A and B were 66% and 80%, respectively.
Figure 2D – This appears to be a metopic ridge, not a sagittal ridge. If the image is kept, it would be helpful to have it rotated 180 degrees. Note, no sagittal ridge is apparent in figure 3.
RESPONSE
Thank you for this comment. Sorry for the mistake. The figure has been corrected has suggested.
Line 139 – It is not specifically stated why exome sequencing was pursued. This paragraph should be rearranged to state the purpose. It should also explicitly state why the parent of origin is important for this report.
RESPONSE
Thank you for this comment. Our lab has used whole exome sequencing for screening the mutations. It is our routine as it is not that expensive and it shows us the pathogenic variants and other variants that may have the effects on the phenotypes of the patients. The relevance of the parental origin of the mutation is described in the discussion as..
FGFR2 mutation and its parental origin
Spontaneous mutations in FGFR2 were usually of paternal origin and associated with advanced paternal age. This is because FGF/FGFR signaling has a crucial role in initi-ation, maintenance of spermatogenesis and clonal expansion of spermatogonial cells expressing proteins with gain-of-function or oncogenic properties [21,28,29]. The father and mother of our patients were 26 and 25 years old, respectively, at the time of con-ception. The mutation found in our patients were of paternal origin, suggesting that the mutational event could have taken place during spermatogenesis [28], even though the father was quite young.
Line 184 – The existence of another report of presumed monozygotic twins with Pfeiffer syndrome should be added to the introduction.
RESPONSE
Thank you for this comment. We understand your point. However, clinically it cannot be certain that those twins had Pfeiffer syndrome. This is because the pictures of the broad thumbs and great toes, which are the characteristic features of Pfeiffer syndrome, were not illustrated. Authors did not report on the causative mutation and the facial features of patients resembling those of Crouzon syndrome. Please kindly allow us to only discuss this in the discussion part.
Line 208 – Typographical error “Weiss”
RESPONSE
We believe Jackson-Weiss is correct.
Line 260 – This sentence is unrelated to upper airway obstruction.
RESPONSE
Thank you for this valuable comment. We agree with your comment. We added the increased intracranial pressure in the subheading. The subheading now is …
Increased intracranial pressure, upper airway obstruction, and sleep apnea
keep the title as it is.
These responses are also in the uploaded file

Reviewer 2 Report
This manuscript reports some interesting findings. As a reviewer, I would recommend to accept this manuscript.
Author Response
Response to comments of Reviewer 2
This manuscript reports some interesting findings. As a reviewer, I would recommend to accept this manuscript.
RESPONSE
We are thankful to Reviewer 2 for seeing value in our work and for your kind words.

Reviewer 3 Report
Nicely presented manuscript, based on a case report. However, the title 'Clinical and genetic studies of the first monozygotic twins with 2 Pfeiffer syndrome' is not representative of that manuscript. You cannot support the term ' clinical and genetic studies' based on a case report. Mreover, the fact that this is this is the first report of monozygotic twins with Pfeiffer syndrome is not adequate to submit a manuscript under the term 'article'
Author Response
Response to comments of Reviewer 3
Nicely presented manuscript, based on a case report. However, the title 'Clinical and genetic studies of the first monozygotic twins with 2 Pfeiffer syndrome' is not representative of that manuscript. You cannot support the term ' clinical and genetic studies' based on a case report. Moreover, the fact that this is this is the first report of monozygotic twins with Pfeiffer syndrome is not adequate to submit a manuscript under the term 'article'
RESPONSE
Thank you for your comment. However, since we studied extensively the clinical and genetic aspects of the first monozygotic twins with Pfeiffer syndrome. Please kindly allow us to keep the title as it is.

Round 2
Reviewer 1 Report
The authors adequately answered concerns raised in the initial review.
Author Response
Comment by Reviewer 1
The authors adequately answered concerns raised in the initial review.
RESPONSE
Thank you so much for your kind words.
Reviewer 3 Report
I have read your reply to my comments. Even though I insist on my comments, I would like to mention that this manuscript presents a case with peculiar and interest findings. Based on that, I could suggest this manuscript for publication.
Author Response
I have read your reply to my comments. Even though I insist on my comments, I would like to mention that this manuscript presents a case with peculiar and interest findings. Based on that, I could suggest this manuscript for publication.
RESPONSE
Thank you for your kind comments. This revised manuscript has been edited by a native speaker.
